# Subjective Emotional Well-Being, Emotional Intelligence, and Mood of Gifted vs. Unidentified Students: A Relationship Model

**DOI:** 10.3390/ijerph16183266

**Published:** 2019-09-05

**Authors:** Ana María Casino-García, Josefa García-Pérez, Lucía Inmaculada Llinares-Insa

**Affiliations:** 1Departamento de Educación Inclusiva y Desarrollo Sociocomunitario, Universidad Católica de Valencia San Vicente Mártir, C/ Sagrado Corazón, 5, Godella, 46110 Valencia, Spain; 2Department de Psicologia Social, Universitat de València, Avda. Blasco Ibáñez, 21, 46010 Valencia, Spain

**Keywords:** gifted, emotional intelligence, subjective well-being, mood, unidentified students, healthy, adolescence, child

## Abstract

Subjective well-being (SWB) is a basic component of the health of children and adolescents. Studies of SWB in gifted students are scarce and show contradictory results. Some researchers consider these groups to be vulnerable, and according to some reports they are more often involved in situations of harassment as victims and/or harassers. Emotional intelligence (EI) is related to SWB and can be a protective factor in these situations. However, the underlying mechanism remains relatively unexplored, especially in the affective dimension of SWB. The present study develops and tests a model for the mediating role of mood in the relationship between EI and SWB. The participants were 273 Spanish students aged 8 to 18 years, distributed into two samples: sample 1, gifted students, and sample 2, unidentified students. The results showed that (1) gifted students exhibited lower scores in EI (specifically, in clarity) and SWB (specifically, in positive experiences) and higher scores on the sadness dimension of mood states and that (2) EI was positively related to SWB, and mood was a significant mediator in the relationship between EI and SWB. The mediating role of the positive mood is given in both groups; however, the negative mood only mediates this relationship in gifted students. The results are discussed, theoretical and practical contributions to the literature are proposed, and implications for parents and teachers are suggested.

## 1. Introduction

Subjective well-being (SWB) is a basic component of health that predicts emotional, cognitive, and behavioral engagement, as well as academic performance [1]. SWB predicts student engagement [2]. High levels of well-being can also play a protective role against victimization among peers in students with high levels of psychopathology [3].

SWB is particularly important in childhood and adolescence, as the foundation for mental health is being laid, and school is an important means to promote SWB [4]. These are all issues of importance for gifted children and adolescents and their families [5].

A significant percentage of gifted students are victims of bullying and cyberbullying [6,7]. Being different in the eyes of peers increases the likelihood of becoming a victim. Gifted children can be seen as different since they are intellectually more advanced than their classmates and can claim the attention of teachers [7]. In American society, there is a cultural tradition of anti-intellectualism that often reflects the circumstances of the classrooms [6]. High achievement can be valued by parents and teachers, but not by the general school culture [8]. In addition, these children may experience asynchronous development in social and emotional areas [9]. Bullies may include classmates but also adults (e.g., parents and teachers). Studies point to gender differences; men are harassed to a greater extent than women [8], but gifted females have more peer victim levels [10]. The strong awareness of justice presented by gifted children can lead them to not understand why something unfair is happening to them if they have done nothing wrong [11], and they may feel confused and distressed [12]. They can live as failures, being unable to solve their problems without the help of their parents or teachers [11] or have anxiety when thinking they are causing them problems and worries [12]. These children are also at risk of becoming bullies in an attempt to regain control [8].

In Spain, the achievement results in international studies, in comparison with other countries, point to an excessive number of students with low achievement and a small number of students with high achievement [13]. The number of students identified as high achievers is very low (0.2%) in Spain [14]. The current Education Act [15] considers that these students may present specific needs for educational support and establishes enrichment and acceleration as measures for their attention. To activate these measures, a favorable report from the authorized School Guidance Service is required. In this sense, in the Valencian Community there are only 94 registered gifted students, and no data for later years [14]. In addition, teachers have little knowledge on the subject, even if they have completed specific training on giftedness [16]. This means that students are not sufficiently stimulated at school and see their underdeveloped potential [13]. As in other countries, gifted students scored higher than the general population on levels of bullying and cyberbullying. A total of 55.1% of gifted students are victims, but if the victim status is added to that of the aggressor, the total is 83.2% of the sample [17]. Specifically, González-Cabrera et al. [18] showed that 31.5% were related to the cyber victim profile and 10.6% to the cyberbully profile in a Spanish sample. Almost 25% of respondents believe that, in some way, teachers had fostered them to be victims. Researchers have found a psychological impact in cases of victimization (e.g., higher scores of depression, stress, and anxiety, as well as lower perceived quality of life) [17].

The presence of mental pathology associated with gifted individuals is an issue that has generated much controversy. It seems that there are no significant differences between this group and unidentified children. Although they may occasionally score slightly higher on mental health measures [19], gifted students do not often perceive themselves as successful or happy people [20].

Much controversy also exists about whether gifted children experience lower or higher levels of well-being than their unidentified peers. Research on well-being in gifted children and adolescents shows contradictory results. The combination of different factors, such as the giftedness type, educational adjustment, and personal characteristics of each child, could contribute to the heterogeneity of the results obtained [21].

Among the personal factors, we consider emotional intelligence (EI) to be relevant. It may play a fundamental role in the academic success of students and in their ability to adjust in the classroom, and it may also become a protective factor in adaptation processes and against aggression, for example [22,23]. The literature shows that EI is associated with positive mood [24].

A better understanding of the emotional profile of gifted students is essential to understanding the factors that affect their well-being. Few studies have analyzed the role of EI in gifted students, and the results have been varied and inconclusive [25]; thus, more research is needed [22]. Therefore, it is the objective of this work to analyze SWB and EI, considering the contribution of both to mood.

### 1.1. Subjective Well-Being in Gifted Children and Adolescents

Studying the well-being of gifted students and understanding what improves it is important for their academic and personal development [5]. The literature distinguishes between psychological well-being (PWB) and SWB. PWB is a desirable quality, judged by external observers who apply certain criteria (virtue, sanctity, success, etc.) [26]. SWB is different for each individual, depending on his or her experiences and not on objective conditions (health, age, etc.). SWB involves a global assessment of all aspects of a person’s life in a specific time period; it has three components: a cognitive dimension (judgments of life satisfaction) and two affective dimensions (high levels of positive affect (PA) and low levels of negative affect (NA)).

There is long-standing interest in how talent affects well-being. Diener [26] assumed that as intelligence was a highly valued resource in society, it would be a personality variable that would be strongly related to SWB. However, very few studies analyze well-being in gifted children [25], and these studies offer contradictory results.

It seems evident that giftedness influences well-being. Some authors consider giftedness as a mechanism that promotes individual social adaptation; they highlight the social value of talent [26] and that these individuals better understand each other and have better strategies to resolve conflicts, have a better academic self-concept [20], etc. However, other authors consider that giftedness is a risk factor that increases vulnerability—gifted subjects may be stigmatized [27]. There are negative stereotypes towards giftedness, especially regarding socio-emotional characteristics [28]—they feel more alone [29]; they are more sensitive to interpersonal conflicts; and they are exposed to more stressors, as the environment sometimes exerts great pressure on them [22]. They may be afraid of failure or develop unhealthy perfectionism [30,31]. They may have more critical attitudes towards the world [21] and experience emotional dyssynchrony or overexcitability [9]. This would be especially true in twice-exceptional (2e) students, who have some learning or emotional disabilities in addition to being gifted [22].

In school contexts, many gifted children do not receive an educational response tailored to their needs. A lack of support and lack of an educational response can lead to the emergence of disruptive behaviors, including challenging teachers’ authority, boredom, and difficulties in relationships with their peers and family members [32]. Negative behaviors from parents and teachers, information that is communicated to children and pressure from academically demanding contexts (big-fish-little-pond effect) are associated with an increase in fear and anxiety [22]. Recognition and support from teachers, as well as peer support, are positively related to students’ well-being, even when added difficulties exist [33]. In contrast, being labeled as a gifted student, especially in environments where there are negative social representations towards study, may be associated with negative emotional outcomes and may lead them to hide their talents [27].

Most studies of gifted children and adolescents concerning WB have analyzed indicators of PWB (self-esteem, self-concept, internalizing and externalizing disorders, etc.) (e.g., [34]). There are fewer studies on SWB; of them, most studies on SWB focus on the analysis of the cognitive component satisfaction with life, for example, in relation to certain aspects such as perfectionism [31]. When comparing the judgments of gifted subjects and those not identified, no significant differences are found between the two groups (e.g., [35]). However, assessments of gifted students seem to be more strongly linked to their school experiences, while in the group of unidentified students, school satisfaction only slightly contributes to overall life satisfaction. This indicates that gifted students may make well-being judgments in a different way [36].

In this sense, some researchers have studied positive and negative perfectionism and found that these factors differentially correlated with the different components of SWB [30]. Positive perfectionism correlated with satisfaction with life and PA, while negative perfectionism was fundamentally associated with NA.

Regarding the study of PA and NA, Chen, Fan, Cheung, and Wu [37] consider that PA contributes to SWB, while NA and peers’ alienation are negatively related to life satisfaction. We have not found studies that analyze the affect dimension in gifted students (e.g., [35]). This leads us to include SWB in the model, considering PA and NA.

There are few gifted students identified in Spain [14]. These children are regarded as different and stigmatized [7]. In addition, there is no adjustment between their characteristics and the educational responses they receive [13], they are more involved in situations of bullying and cyberbullying [18], and most do not feel supported by their teachers [17]. This may be the cause of higher levels of NA and lower levels of AP and, therefore, lower SWB [25]. This allows us to formulate the first hypothesis.

**Hypothesis 1 (H1).** *Gifted children will have lower SWB than unidentified children*.

### 1.2. Emotional Intelligence in Gifted Children and Adolescents

Salovey and Mayer proposed EI as a human’s ability to reason about emotions. The authors set out with two assumptions: (a) intelligence is the ability to carry out abstract reasoning, and (b) intelligence can be considered a system of mental abilities. Then, emotionally intelligent people (a) perceive emotions accurately, (b) use emotions to facilitate thinking, (c) understand emotions and emotional meanings, and d) effectively regulate and manage emotions in themselves and others [38].

EI has been shown to be important to well-being [39]. It may be a health predictor; in particular, EI correlates positively with mental health and negatively with depression (e.g., [40]). Young people with high EI have more positive social relationships and are less likely to engage in risky behaviors [41]. Emotional skills predict a wide range of outcomes related to adaptation, and coping with stress and well-being act as protective factors against aggression [23]. Thus, the perception, understanding, and management of emotions can play a role in the academic success of students [42] and their inclusion in the classroom [22].

There is a solid theoretical basis for suggesting that cognitive ability and EI are moderately correlated. However, the empirical data suggests very low and inconsistent associations between EI traits and cognitive capacity [41]. Research on the EI of gifted children and adolescents is an important topic for counselors, teachers, and parents [25] but studies on this topic are scarce [43] and offer mixed results. Some research finds that gifted students have higher levels of EI (e.g., [44]); others offer equivalent results to those of unidentified students (e.g., [45]). When analyzing components with similar overall scores, gifted students have higher levels of adaptability but lower levels of stress management and impulse control capacity [46], which can make them vulnerable.

The results also appear to depend on the measure used. Zeidner, Shani-Zinovich, Matthews, and Roberts [47] examined the emotional self-perception of young gifted people aged 13 to 15 years old compared to their actual ability. Interestingly, young gifted people perceived themselves to have less emotional competence, even although they obtained better results in the skills test. These students attended classes for gifted pupils; the authors attributed the differences to social comparison processes. Children could perceive themselves as being separated from normal social interactions due to being gifted. Another explanation might be that gifted children are better acquainted with their personal limitations, while those who are not gifted overestimate their qualities in a naive way.

In principle, accepting the independence between EI and cognitive ability and knowing that EI measures have not been systematically included when identifying gifted students in Spain, we propose the second hypothesis in our study.

**Hypothesis 2 (H2).** *The EI of gifted children and adolescents will not differ from that of unidentified children*.

### 1.3. Subjective Well-Being, Emotional Intelligence, and Mood in Gifted/Unidentified Children and Adolescents

EI is conceived as an indicator of psychological adjustment and a key precursor of feelings associated with well-being. People with higher EI scores have broader support networks, maintain closer social relationships, and deal more easily and adaptively handle the stressors and nuisances of everyday life [48].

Scientific research indicates that there is a positive relationship between EI and SWB, regardless of the theoretical currents and the measuring instruments used [39]. Therefore, we put forward hypothesis 3.

**Hypothesis 3 (H3).** *EI positively predicts SWB*.

SWB may be conceived as a balance of both PA and NA, having a momentary state and reflecting the emotional status quo at a specific time. As these momentary states accumulate over time, they add up and begin to reflect a central tendency or characteristic level of emotional well-being around which the person fluctuates [49]. The average levels are the result of two components: the frequency and the intensity of affect. There are personality differences in affective intensity; however, studies also stress the importance of life circumstances [50].

Moods refer to global affective states without a specific cause or motive or a clear beginning [51]. Moods are affected by various personal variables, such as physical health or coping, and social variables, such as peer relationships or social support [52]. Pleasant moods tend to bias information processing towards more positive assessments; a similar pattern occurs for negative moods.

In current studies, the emotional component, understood as the proportion of PA to NA, correlates with other measures of general well-being [49]. Satisfaction with life correlates moderately with PA levels and inversely with NA levels and may be influenced by contextual factors, but life satisfaction is relatively independent and difficult to change [53]. However, the emotional components of SWB are more reactive to situational influences and are easier to modify [54]. Thus, we have chosen them as the object of study.

Emotions in adolescents and young adults are particularly important. Very few studies have examined the evolution and differences related to feelings and positive and negative experiences in educational contexts. However, during adolescence, there is a decrease in positive feelings and an increase in negative feelings [4].

If the previous model is assumed, one way to improve the emotional balance of the person and, therefore, his or her general well-being and health, in general, would be to increase the numerator of the ratio (PA) or decrease the denominator (NA) of the equation [49]. Most people, in all cultures, tend to be moderately happy and are able to overcome the bias of the hedonic treadmill and maintain at least a minimum well being (WB). The broaden-and-build theory argues that positive emotions initiate upward spirals of well-being improvement [55]. It is also likely that EI refers largely to the capacity to handle NA.

However, the contributions of PA and NA to WB are not equal in magnitude. Pollyanna’s principle says that there is a broad tendency to process pleasurable information more accurately and efficiently than unpleasant or neutral information [56]. Negative emotions and moods last longer and have larger impacts than positive ones [49].

EI could reduce the occurrence and duration of negative emotions and mood changes and could increase the frequency and maintenance of positive emotions over time [48]. EI has been positively associated with optimistic mood and acts as a buffer against stressing factors [24]. EI also correlates with SWB [39].

There is also an association between perceived EI and the affective component of SWB, but this association is smaller than that between EI and the cognitive component of SWB. Some authors explain this phenomenon by referring to the greater temporal stability of self-reported emotional abilities and satisfaction with life rates (which are not moldable in short periods) than that of most modifiable daily moods assessed by an affective index. Sánchez-Álvarez et al. [39] also indicate the desirability of developing research focused on the relationship between EI and the affective component of SWB. Moreover, Fernández-Berrocal and Extremera [40] say that the mediators of the relationship between EI and health-related outcomes should be sought.

Several authors have found a relationship between emotional awareness, moods and adjustment indicators in preadolescence and adolescence. In this sense, emotional consciousness is reinforced by moods. When there are few emotional competence skills, there is a tendency to generate a negative mood, which leads to high levels of prolonged stress [57]. In this sense, we propose that moods will mediate the relationship between EI and the emotional component of SWB (the balance between AP and NA). Therefore, we put forward hypothesis 4.

**Hypothesis 4 (H4).** *Moods will mediate the relationship between EI and SWB*.

Martin-Krumm et al. [4] point to the importance of evaluating WB in adolescents with unique or vulnerable characteristics; however, few studies have analyzed the affective dimension of SWB in gifted children or adolescents.

Vialle et al. [29] found that gifted children felt more alone. Compared to their peers, they had higher average NA and lower average PA, obtaining a worse result; no significant differences were found between gifted students and their peers except in the case of sadness. The fact that many gifted children do not receive an educational response tailored to their needs, coupled with their involvement in harassment situations, leads us to believe that, in Spain, this population may suffer negative emotions more often than unidentified students.

At the academic level, activities are considered pleasant when the challenge is adjusted to the person’s skill level. If an activity is too easy, boredom will occur; if the activity is too difficult, it will generate anxiety. When a person is involved in an activity that requires intense concentration and where the person’s abilities and the task challenge are approximately equal, a pleasurable flow experience will occur. Student interest in academic activity is associated with positive emotions; intrinsic motivation and global perceived control over progress levels are highly correlated with WB [58]. The “flow” of academic commitment and positive relationships with peers who have similar skills and motivations may protect the gifted student from the stress of academic requirements and may prevent them from experiencing worse adjustment [59]. According to a study conducted by Al-Onizat [45], there is a positive relationship between social adaptation to school and EI ability, and the closest relationship appears in the relationship with peers. The relationship between all dimensions of EI and all dimensions of the school social adaptation scale is also positive, and the closest relationship occurs between the relationship with peers and general mood. From this information, hypothetical mediation (H4) will be tested separately for gifted and unidentified students. Then, in gifted students, positive and negative mood mediates the effect of EI on SWB and in unidentified students, positive mood mediates the influence of EI on SWB.

## 2. Materials and Methods

### 2.1. Participants and Procedure

The study design was cross-sectional and we used the convenience sampling technique [60] based on participants who are easily accessible, but we deliberately chose certain people based on their characteristics. We used this technique because it was necessary to select an equivalent number of gifted and unidentified students with similar characteristics. The criteria used to collect both samples were: easy accessibility, geographical proximity, availability at a given time, and willingness to participate. Moreover, in Spain, the percentage of gifted boys and girls is different; the Ministerio de Educación y Formación Profesional [14] showed that there are more boys identified as gifted (approximately 70%). There are fewer girls because they hide their abilities, and in the classroom, they are called invisible girls in the classroom [61]. The conditions for participating in the study were that the students must be identified or unidentified as gifted and be in compulsory education. Data were collected through self-report questionnaires completed voluntarily by the participants in the presence of the researcher after they had provided their own and their parents’ informed consent. Participants received instructions and information about the procedure for filling out the questionnaire, and the researcher helped to explain that there were no right or wrong answers and was present to resolve any doubts that arose [62]. Exact response rate could not be determined. There were missing data on conflictive items representing approximately 0.2%, and these participants were excluded from the analyses. This study was carried out in accordance with the ethical guidelines of the American Psychological Association and the Declaration of Helsinki, and it received approval from the Ethics Committee of the Catholic University of Valencia (UCV/2015-2016/05).

A total of 273 Spanish children in two samples were surveyed (gifted students and unidentified students). Sample 1 consisted of 132 gifted Spanish students (27.3% female and 72.7% male) from the region of Valencia. The mean age was 10.54 years old (SD = 2.38). The majority of the sample was studying in primary school (71.2%), and 26.5% were completing compulsory secondary education. Sample 2 consisted of 141 unidentified Spanish children (32.6% female and 67.4% male) from the region of Valencia. The mean age was 10.59 years old (SD = 2.40). The majority of the sample was studying at primary school (68.8%), and 26.2% were completing compulsory secondary education.

This study has been carried out in accordance with the ethical guidelines of the American Psychological Association and the Helsinki Declaration and has been approved by the Ethics Committee of the Catholic University of Valencia (UCV/2015-2016/05).

### 2.2. Measures

The following questionnaires were used.

The Scale of Positive and Negative Experience (SPANE) [63]. The Cassaretto and Martínez [64] Spanish adaptation was used. This scale consists of 12 items rated on a five-point Likert-type scale (1 = very rarely or never to 5 = very often or always). This scale has a unifactorial structure of balanced scores for positive and negative feelings. An example of a positive item is “Good.” An example of a negative item is “Angry”. The scale shows high internal consistency (Sample 1 α = 0.88; Sample 2 α = 0.83).

Trait Meta-Mood Scale-24 (TMMS-24) [65]. The Salguero, Fernández-Berrocal, Balluerka, and Aritzeta [66] Spanish adaptation were used and validated. TMMS is used to understand perceived EI, and it includes beliefs and attitudes about emotional experience. This scale has 24 items, and participants were asked to rate each item on how accurately it represented the actions they took towards another person, using a rating scale ranging from 1 (strongly disagree) to 5 (strongly agree). It examines the meta-knowledge of emotions and personal skills regarding attention to, clarity about and repair of emotional status. Examples of items are: “I pay a lot of attention to how I feel”, “I almost always know exactly how I am feeling” and “No matter how badly I feel, I try to think about pleasant things”. The scale shows high reliability (Sample 1 α = 0.85; Sample 2 α = 0.85).

Mood Questionnaire [51]. In this study, we used an adaptation of the questionnaire from Górriz et al. [52] and their validation. This scale consists of 20 items on a three-point Likert-type scale (1 = Never, 2 = Sometimes and 3 = Often). It is composed of four factors: Fear (e.g., “I feel frightened”), Sadness (e.g., “I feel miserable”), Happiness (e.g., “I feel cheered”) and Anger (e.g., “I feel angry”). This study used two specific dimensions, however: positive (happiness) and negative mood (sadness, fear, and anger). The scale shows high reliability for the positive dimension (Sample 1 α = 0.80; Sample 3 α = 0.81) and the negative dimension (Sample 1 α = 0.86; Sample 2 α = 0.80).

Moreover, two control variables were measured: sex and educational level (primary, compulsory secondary education, vocational training, and higher education).

### 2.3. Data Analysis

First, descriptive statistics (the mean and standard deviation) were calculated to analyze the EI, mood, and SWB of gifted students. Second, we used the t-statistic to identify significant differences from the average and correlations. Third, the data were analyzed using structural equation modeling (SEM) techniques to test the model proposed above. To test the hypotheses, path analysis was then conducted using the ML estimation method. Given the small sample size and the number of parameters to be estimated, we modeled relationships among observed (not latent) variables. We tested a moderated model that included all the study hypotheses. This model presents the mediating effect of positive/negative mood in the relationship between emotional intelligence and well-being. The path analysis approach allows the hypothesized relationships to be examined. This analytical technique facilitates the investigation of direct and indirect effects between variables. The indirect effects involved in the model were tested using the bias-corrected (BC) bootstrap confidence interval method. We tested the hypothesized relationship using SEM in the total sample. Later, multigroup SEM was employed to separately test the model in gifted students and unidentified students. Then, SEM was carried out in gifted students and unidentified students separately. Maximum likelihood (ML) parameter estimates were calculated and checked for robust methods. Model fit was assessed with a combination of fit indices. A nonsignificant chi-square statistic indicated good model fit; however, the chi-square test is sensitive to sample size. Since 1990, recommended cutoff values were indices of ≥0.90, but it has since been shown that the cutoffs recommended by Hu and Bentler are not universally applicable. Moreover, parameter estimates are influenced by sample size. If the initial model offers a poor fit to the data, the second step is to modify the model.

Furthermore, we used analytical techniques to analyze direct and indirect effects between variables. The indirect effects involved in the model were tested using the method proposed by MacKinnon et al. [67]. Additionally, we analyzed effect size for mediation through complete or partial mediation.

Combinations of descriptive and inferential statistics were calculated with the Statistical Package for Social the Sciences (SPSS) version 24, SEM was performed using Analysis of Moment Structures (AMOS) software version 24 (IBM SPSS Amos 24, Wexford, USA), and PROCESS by Hayes (Columbus, USA). Then, 2000 bootstrap resampling was performed with confidence intervals set at 95%. The sample size was adequate, the correlations between variables were not high (i.e., 0.85), and the sample data did not follow a standard normal distribution.

## 3. Results

### 3.1. Descriptive and Preliminary Analysis

The means and standard deviations, as well as Pearson bivariate correlations, for the studied variables, were calculated (Table 1 and Table 2). There were moderate positive correlations between EI, SWB, and positive/negative mood. Higher levels of negative mood were also found to be concurrently associated with lower EI and well-being; however, there were no statistically significant correlations between EI and negative mood in unidentified students.

Moreover, we conducted an independent *t*-test based on the gifted or unidentified group. The results are shown in Table 1. Our descriptive analysis results of the descriptive analysis show that students had medium scores in some variables. There were high scores of positive experiences and lower scores of negative experiences and mood. There were significant differences in EI (t = 2.19; *p* < 0.02) and SWB (t = 1.93; *p* < 0.05) between gifted and unidentified students. There were significant differences in clarity (t = 2.75; *p* < 0.01) and positive experiences (t = 2.36; *p* < 0.01), specifically. In both cases, unidentified students had higher scores; however, gifted students significantly feel sadness (t = −2.10; *p* < 0.03). These results provide support for H1, but H2 is not confirmed. Moreover, in the two samples in this research, there were no statistically significant differences in the study variables based on sex and educational level. Therefore, these control variables were not included in the model.

### 3.2. Test of the Hypothesized Model

To test the hypotheses, path analysis was then conducted using the ML estimation method. Given the sample size and the number of parameters to be estimated, we modeled relationships among observed (not latent) variables. We tested a moderated model that included all the study hypotheses. This model proposed the mediating effect of mood in the relationship between EI and SWB. The path analysis approach allows the hypothesized relationships to be examined. This analytical technique facilitates the investigation of direct and indirect effects between variables. The indirect effects involved in the model were tested using the bias-corrected (BC) bootstrap confidence interval method. First, we tested the hypothesized model for all students. Table 3 presents the goodness of fit indices. The model presents an acceptable fit to the data, and the estimators are significant in the total sample. To further verify mood’s mediation effect, PROCESS was used and we have computed the mediator’s BC confidence interval. EI is significantly related to positive (Coefficient = 0.2, SE = 0.03, *p* = 0.001) and negative mood (Coefficient = −0.1, SE = 0.03, *p* = 0.001). Positive (Coefficient = 1.65, SE = 0.17, *p* = 0.001) and negative mood (Coefficient = −1.66, SE = 0.17, *p* = 0.001) is significantly related to SWB after controlling for EI. The analysis of the indirect effect of EI SWB, through the positive (indirect effect = 0.32; 95% BC confidence intervals being [0.19, 0.49]) and negative mood (indirect effect = 0.16; 95% BC confidence intervals being [0.05, 0.27]), showed a statistically significant mediation. The direct effect is statistically significant (Coefficient = 0.26, SE = 0.08, *p* = 0.01); thus, we could conclude that positive and negative mood partially mediate the relationship between EI and SWB. These results provide support for H3: EI predicts well-being. The hypothesized mediating role of positive and negative mood in the relationship between EI and SWB was tested, and the results are in support of H4 (Figure 1).

Later, we examined whether there were differences between gifted and unidentified students. Multigroup analysis showed the model fit adequately (Table 4). The results, however, revealed that the relationship between EI and negative mood was not significant in the unidentified students. In the gifted group, there was no significant relationship between EI and SWB. Table 3 shows the standardized coefficients for the impact of the model on SWB. This result supports H4: negative and positive mood mediates the relationship between EI and SWB differently in gifted and unidentified students. Thus, we did not estimate whether the model was identical or equivalent to the constrained model across groups.

Finally, we analyzed the model in the gifted and unidentified gifted students separately. Two re-specified models were estimated for gifted students and unidentified students (Figure 2). The results showed that the re-specified models were valid for the two groups. Table 3 shows the goodness of fit indices. The estimators also indicate that all the estimated variables correlate adequately, as well. In the re-specified unidentified student model, EI was significantly related to positive (Coefficient = 0.18, SE = 0.04, *p* = 0.001) but it was not to negative mood (Coefficient = −0.04, SE = 0.04, *p* = 0.27). Both positive mood (Coefficient = 1.43, SE = 0.24, *p* = 0.001) and negative mood (Coefficient = −1.65, SE = 0.25, *p* = 0.001) are significantly related to SWB after controlling for EI. The analysis of the indirect effect of EI SWB, through the positive mood (indirect effect = 0.26; 95% BC confidence intervals being [0.09, 0.46]) showed a statistically significant mediation. The direct effect is statistically significant (Coefficient = 0.32, SE = 0.12, *p* = 0.006); thus, we could conclude that positive mood partially mediates the relationship between EI and SWB.

In the re-specified gifted students model EI was significantly related to positive (Coefficient = 0.20, SE = 0.05, *p* = 0.001) and negative mood (Coefficient = −0.14, SE = 0.05, *p* = 0.01). Both positive mood (Coefficient = 1.94, SE = 0.24, *p* = 0.001) and negative mood (Coefficient = −1.73, SE = 0.23, *p* = 0.001) are significantly related to SWB after controlling for EI. The analysis of the indirect effect of EI on SWB by positive mood (indirect effect = 0.37; 95% BC confidence intervals being [0.18, 0.63]) and negative mood (indirect effect = 0.24; 95% BC confidence intervals being [0.08, 0.41]) showed a statistically significant mediation. The direct effect was not statistically significant (Coefficient = 0.18, SE = 0.13, *p* = 0.18); thus, we could conclude that positive and negative mood totally mediates the relationship between EI and SWB. Therefore, positive and negative mood mediates the relationship between EI and SWB in gifted students, but in unidentified students, only positive emotions mediate this relationship.

## 4. Discussion

Well-being is a key factor that determines people’s health. In this research, we focused on SWB, specifically on the emotional balance each individual finds based on their experiences, comparing PA and NA. In childhood and adolescence, the foundation for mental health is laid. WB is especially important at this stage, when negative emotions increase and positive emotions decrease [4]. A high level of SWB could play a protective role against victimization [3].

The literature on the subject topic shows that EI is closely related to SWB [39]. EI has been positively associated with optimistic moods and has been shown to act as a buffer against stressors [24]. Fernández-Berrocal and Extremera [40] mentioned the need to analyze mediators of the relationship between EI and health-related outcomes. Thus, this study investigates the mediating role of mood in the relationship between EI and SWB. In addition, we do not know of any studies that have investigated this issue in gifted children, adolescents, and unidentified groups. The present study considers giftedness as an essential personal factor that has an impact on well-being and psychosocial adjustment. All these issues are of importance for gifted children and adolescents and their families [5], as well as for the professionals working with them: teachers, psychologists, doctors, etc. [25].

Our results show that there are significant differences in SWB and EI between gifted students and unidentified students. Thus, H1, but not H2, is verified. Our H1 results do not match those of Bergold et al. [35], who found no significant differences between gifted and unidentified students in terms of in satisfaction with life; moreover, they consider that giftedness is not a risk factor for the deterioration of psychosocial well-being. However, our study differs in two important aspects: (1) Bergold et al. evaluated the cognitive component of SWB, and we evaluated the affective component (as recommended by the authors themselves in their study), and (2) their students did not know whether they belonged to the gifted group or not. Regarding the first aspect, Chen et al. [37] found that PA contributes to SWB, while NA and the alienation of peers, depression, and anxiety are negatively related to life satisfaction. In Spain, the percentage of gifted students in bullying situations is very high, and a student experiencing some type of victimization may have higher scores of depression, stress, and anxiety. In addition, almost 25% of students believe that their teachers have led them into such a situation [17]. School experiences are key to the well-being of children and adolescents [68]. In addition, school satisfaction is closely related to overall satisfaction with life in gifted students [36]. Authors such as Kroesbergen et al. [69] report that this decline in well-being may also be due to a lack of support from teachers. It must be added that scarce training for teachers [16], which can make adjusting the educational response difficult, can cause boredom [70] and can decrease the “flow” of their academic commitment and positive relationships with peers [59].

Because intrinsic motivation and the overall perceived level of control over progress are highly correlated with SWB [58], low student motivation due to a lack of adjustment in the educational response, a feeling of lacking control and the absence of teacher support could explain the results obtained, even if the student was not being harassed. Regarding the second difference, whereas the students in the Bergold et al. [35] study were unaware of their ability, our gifted students were labeled, and many of them belonged to gifted associations. In Spain, there is a very low incidence of identified students, –0.2%, according to the Ministry of Education, Culture, and Sport [14]—thus, being gifted labels one as ‘different’. Gifted students do not care that others know that they value education and work to do well in school, but they do not want others to truly perceive the extent and nature of their differences [27], and in our system, they are labeled. There is no national detection protocol, nor are there common criteria in the development of existing ones.

Although an identification procedure has recently been published in our autonomous community [71], the evaluation is carried out on-demand, and its application in a given course is not compulsory to promote detection. By not identifying potential needs, the educational response also cannot be adjusted. Because there is no generalized protocol, many students are identified because they are profoundly gifted or because they present double exceptionality, and these groups may encounter more socio-emotional challenges [43]. Moreover, social representation and unfavorable stereotypes also occur in Spain, where the number of students achieving excellence is low (LOMCE, Preamble V [15]); in particular, these students are considered to have negative socio-emotional characteristics, which logically affects the way their peers and teachers behave around them [28]. Therefore, our results are in line with those of Vialle et al. [29], who found that compared to their peers, gifted students felt more sadness.

For H2, although it is not verified, our results are in line with those obtained by Zeidner et al. [47]. The EI of gifted students is lower than that of unidentified students. In accordance with these authors, the differences found may be due to two reasons: (1) the personal knowledge that gifted students have of their abilities and limitations is more realistic than that of the unidentified students, whose beliefs are more naive and optimistic, and (2) the process of social comparison generates a more negative perception due to feeling different and, in a very high percentage of students, feeling rejected by peers, as discussed above. In contrast, gifted students score lower in clarity and have fewer positive experiences. Specifically, with regard to situations of rejection or harassment by peers, the literature points out that a strong sense of injustice leads them to think that they don’t deserve that treatment when they have done nothing to deserve it [11], and not understanding what happened causes them anxiety [12]. According to Diener, Lucas, and Scollon [72], a possible explanation may be that the “features of one’s life that cannot be explained continue to draw attention and, thereby, affect one’s emotions and overall well-being” (p. 312). Thus, we emphasize the importance of developing EI and SWB during childhood and adolescence.

The results also reveal that EI predicts SWB (H3). This result is consistent with previous research [39] and supports H3 in this study. Thus, when a student has high EI, he or she is motivated to feel positive SWB.

However, as noted above, the relationship between EI and SWB is complex. Some authors have found that the association between EI and the cognitive component of SWB is greater than that with the affective component [39]. Therefore, in this research, we studied the affective component of SWB, evaluating the mediating effect of mood in this relationship (H4).

The results of our study reveal that this mediating effect exists, and therefore, the H4 is accepted. Our model is consistent with the results obtained by Extremera and Rey [73], who found that PA and NA play a mediating role in the relationship between EI and life satisfaction. In the present research, we find that the average mood is different in each group (gifted and unidentified students). In this study, the mediating role of positive mood in the relationship between EI and SWB is given in the two groups; however, negative mood only mediates this relationship only in gifted students. Therefore, we could conclude that positive and negative mood totally mediates the relationship between EI and SWB in gifted students only. In unidentified students, there is a partial mediation by positive mood, but none by negative mood.

One possible explanation would be that a positive mood favors Pollyanna’s principle, helping students process pleasurable information more efficiently and accurately and overcoming the bias suggested by the hedonic treadmill theory. In this direction, Sánchez-Álvarez et al. [39] find that positive emotions help improve people’s overall cognitive assessment of their satisfaction with their own lives and, consequently, their SWB. However, our results show that in gifted students, negative mood has a real impact on SWB.

### 4.1. Theoretical and Practical Implications

In line with Martin-Krumm et al. [4], our research has made it possible to advance the knowledge of the affective dimension of SWB in students, children, and adolescents and in gifted students. A second theoretical contribution from our study is related to the suggestion of Sánchez-Álvarez et al. [39], who highlight the importance of continuing to investigate the relationship between EI and SWB, especially the affective component. In addition, the mediating effect of mood in this relationship has been analyzed because Fernández-Berrocal and Extremera [40] highlighted the importance of discovering important mediators between these constructs.

Our results also have significant practical implications. We highlight two important aspects for improving gifted students’ well-being: the role of personal factors (among them, we highlight EI) and that of environmental factors.

Regarding personal factors, it would be advisable to systematically address the emotional education of gifted students to improve the development of adaptive emotional competences (emotion expression, empathy, emotional regulation, etc.) and strong social relationships, foster an adjusted self-concept, alleviate stress or anxiety, and weave more meaningful lives. These programs are especially valuable when gifted children are vulnerable to socio-emotional deficits [41], but they are also necessary for everyone. Better EI levels may directly help to improve students’ decision-making skills [74]. If the characteristic affective and social components of gifted and talented students are not explicitly addressed, this lapse could compromise the growth of their cognitive potential, their social adaptation, and their well-being [43]. Therefore, social and health policies should consider these variables.

With regard to environmental factors, family and school systems have a strong influence on students’ well-being. When emotional problems arise, they are often due to a mismatch between the social and emotional needs of the gifted child and the abilities of the family, social, and educational environments, along with a lack of care for the child [43]. Generally, teachers, counselors, and health professionals are neither properly trained to recognize the emotional needs of gifted students, nor are they sufficiently trained to help develop the socio-emotional competencies of these individuals. Therefore, their qualifications and an individualized analysis of the needs of these students are essential given the heterogeneity of this group [41]. Additionally, it is essential to systematically plan actions organized to promote parental involvement and support, encourage closeness and support from teachers, and foster a sense of affiliation and inclusion with the teenage peers of gifted students [37].

Along with these two types of actions, the broaden-and-build theory suggests that positive emotions broaden an individual’s consciousness and increase mental openness and clarity of thought [55]. Gifted students respond to challenges and environments promoting talent development with positive emotions and a greater sense of well-being. There are several sources that can lead to low achievement in gifted students; a major source of low performance in the classroom is unadapted self-regulation and motivation. To this end, it is essential to take actions to improve the training of the teachers of gifted students so they can adjust their educational response and incorporate challenging methodologies and activities. It is also important to encourage students, in general, to seek excellence, healthy perfectionism and growth attitudes [31].

Positive (that is, less stressful) relationships with peers (possibly associated with the ability to learn with other adolescents who have similar abilities and motivations) can be used to cushion the potentially negative effects of stressors associated with academic requirements. Flow (how an individual feels when fully involved in an activity) captures the complete involvement of an individual at the time and occurs most often when the activity provides a balance between being challenging and appropriate for their level of abilities. In this sense, feedback from the teacher is a basic precondition. Collaboration with universities to provide these challenging experiences could help growth in this direction [37].

### 4.2. Limitations and Future Research

This study has some potential limitations. First, this research is a transversal study that groups gifted and unidentified children and uses self-reported measures. This choice was justified by the nature of the variables considered in this study. These measures may cause common error bias. To mitigate common method variance, in this study, we use different scale formats and different types of Likert scales [62]; however, we believe that future studies should use data from multiple sources.

Second, our findings may have limited generalizability because we use a convenience sample. However, Etikan, Musa, and Alkassim [60] indicate that this sampling method is useful for this type of study. In addition, the criteria for identifying gifted students were not clear. In Spain, there are few identified children; García-Barrera and De la Flor [16] affirm that 95%–99% of gifted Spanish students are not identified. Because we assume that there are more gifted children and adolescents than those currently identified, we have called the unevaluated children in this study “unidentified”. However, we believe that future studies should evaluate the capacity of all students similarly.

Third, a transverse design draws causal inferences; the results of our study show that EI is related to SWB, that this relationship is mediated by mood and that its interaction causes poor and high SWB; however, we cannot rule out other causal interpretations. Fiedler, Schott, and Meiser [75] argued that statistical analyses about mediation study only one or a few mediators and that other variables may help to identify other true mediators. For example, mood is a dynamic process that relates to emotional states, cognition, and so on. In addition, SWB can be determined from multiple causes that could be related and have various levels of explanation. It would, therefore, be interesting to investigate high-, medium-, and low-quality EI and their effects on SWB over time. Research should also analyze environmental, school, and family factors and their relationship to EI, mood, and SWB. Future longitudinal designs would, thus, provide information on the dynamics of this relationship.

Fourth, the topics analyzed by our study (SWB, EI, and mood) are susceptible to cultural differences. However, there are also cultural variations about the concept of gifted children, gifted education, etc.. Literature suggests that culture is one of the most important sources of psychosocial analysis but personality, socio-economic class, education, sex, age, etc., are also of particular relevance. Moreover, some authors report cultural mismatch between teachers and students. Following Chowdhury [76], futures studies should analyze the diversity of peoples and their cultures in the era of globalization.

## 5. Conclusions

To increase SWB, it is necessary to develop EI. Mood plays an important role in mediating the relationship between EI and SWB. In our study, positive and negative mood mediated this relationship in gifted students. Positive mood, but not negative mood, mediated the relationship between EI and SWB in the unidentified group.

This research shows that gifted children experience lower levels of SWB and fewer positive experiences and felt more sadness than unidentified children and adolescents. Moreover, they had lower EI scores, especially related to clarity. Negative mood also mediated the relationship between EI and SWB. This emphasizes the importance of using evaluation measures and developing prevention programs. Additionally, there is a need to increase the knowledge of and tools used by healthcare professionals and educators to detect the emotional needs of gifted children and develop therapeutics and preventive interventions.

## Figures and Tables

**Figure 1 ijerph-16-03266-f001:**
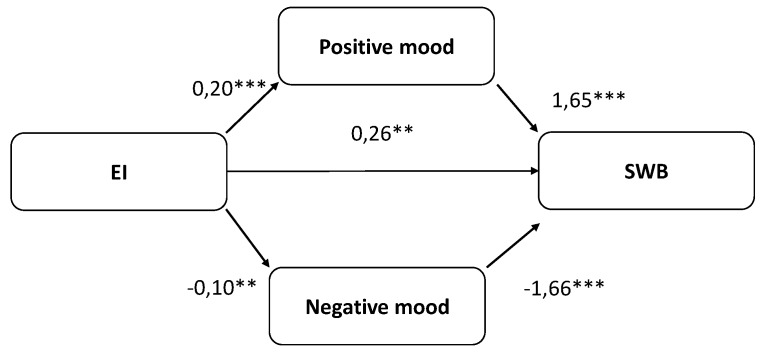
Summary of results of the model in total sample. Non-standardized coefficients are reported. * *p* < 0.05, ** *p* < 0.01, *** *p* < 0.001.

**Figure 2 ijerph-16-03266-f002:**
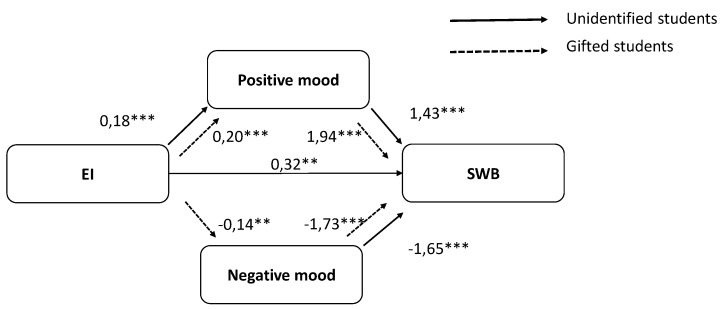
Summary of results of the re-specified model in unidentified children and gifted children. Non-standardized coefficients are reported. * *p* < 0.05, ** *p* < 0.01, *** *p* < 0.001.

**Table 1 ijerph-16-03266-t001:** Descriptive statistics and *t*-test.

Variables	Total Sample		Gifted		Unidentified	
	M	SD	M	SD	M	SD
1. EI	3.42	0.60	3.33	0.61	3.49	0.59
t = 2.19; *p* < 0.02						
Attention	2.96	0.83	2.92	0.87	3.00	0.81
Clarity	3.59	0.89	3.44	0.92	3.73	0.84
t = 2.75; *p* < 0.01						
Repair	3.70	0.77	3.64	0.81	3.75	0.74
2. SWB	2.16	1.26	2.01	1.37	2.30	1.13
t =1.93; *p* < 0.05						
Positive experience	4.23	0.73	4.13	0.77	4.33	0.68
t = 2.36; *p* < 0.01						
Negative experience	2.07	0.73	2.12	0.77	2.02	0.68
3. Positive mood (happiness)	2.79	0.34	2.75	0.35	2.82	0.31
4. Negative mood	1.46	0.32	1.48	0.36	1.44	0.28
Fear	1.47	0.39	1.49	0.41	1.47	0.39
Sadness	1.32	0.35	1.39	0.44	1.27	0.27
t = −2.10; *p* < 0.03						
Anger	1.59	0.45	1.77	0.51	1.58	0.41

Note: EI = Emotional Intelligence, SWB = Subjective Well Being, M = Mean, SD = Standard Deviation.

**Table 2 ijerph-16-03266-t002:** Bivariate correlations (students).

Variables	Total Sample			Gifted			Unidentified		
	2	3	4	2	3	4	2	3	4
1. EI	0.36 **	0.35 **	−0.18 **	0.36 **	0.35 **	−0.25 **	0.34 **	0.34 **	−0.92
2. SWB		0.63 **	−0.60 **		0.65 **	−0.61 **		0.61 **	−0.58 **
3. Positive mood (happiness)			−0.35 **			−0.33 **			−0.36 **
4. Negative mood									

Note: EI = Emotional Intelligence, SWB = Subjective Well Being, * *p* < 0.05; ** *p* < 0.01.

**Table 3 ijerph-16-03266-t003:** Summary of structural equation modeling (SEM) analyses (n = 225).

Model	χ^2^	d.f.	NFI	IFI	CFI
Total sample	27.23	1	0.91	0.91	0.91
Multigroup	28.61	2	0.91	0.91	0.91
Gifted students re-specify	11.66	2	0.93	0.94	0.94
Unidentified Students re-specify	20.04	2	0.86	0.87	0.87

Note. χ^2^ = chi-square; df = degrees of freedom; NFI = Normed Fit Index; IFI = Incremental Fit Index. CFI = Comparative Fit Index.

**Table 4 ijerph-16-03266-t004:** Standardized SEM effects.

Variables	Total Sample			Multigroup Analysis		
		Gifted Students		Unidentified Students		
	β	*p*-Value	β	*p*-value	β	*p*-Value
EI on positive mood	0.20	0.001	0.20	0.001	0.18	0.001
EI on negative mood	−0.10	0.002	−0.14	0.004	−0.04	0.27
Positive mood on SWB	1.65	0.001	1.84	0.001	1.43	0.001
Negative mood on SWB	−1.66	0.001	−1.68	0.001	−1.65	0.001
EI on SWB	0.26	0.003	0.18	0.18	0.33	0.005

Note. EI = Emotional Intelligence, SWB = Subjective Well Being, β non-standardized.

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
