# Peer review of "Subjective Emotional Well-Being, Emotional Intelligence, and Mood of Gifted vs. Unidentified Students: A Relationship Model"

_ijerph, 2019, doi:10.3390/ijerph16183266_

Round 1

Reviewer 1 Report

This paper reports a study on SWB among gifted and unidentified students. Moreover, it tests whether the emotional intelligence is related to SWB both directly and via positive and negative moods. Potential differences between these groups in this hypothesised mechanism is also analyzed.

The focus of this paper is interesting and important from both academic and practical perspectives. The tested model is also relatively well argumented in the literature review. 

I do have, however, some concerns regarding the hypothesis formulation and methodological aspects of the study that would require major revisions. 

First, the formulations of the stated hypotheses are not explicit enough (please more detailed comments below). Second, a more serious, concern is the conducted analysis and the results section. According to the fit statistics the fit between the theoretical model and the data seems poor (here again, please see my more detailed comments on the RMSEA below). The results section should also review the findings in more structured and explicit way.

In addition, the Author(s) could reflect a bit more Spain as a national context. Is the some national characteristics that would explain the findings be generalised to other national contexts as well?  

There are also some problem with the structure of the ms. and it would benefit from a thorough language check. 

More detailed comments:

Abstract:

“Some researchers consider these groups to be vulnerable, and there are studies demonstrating that bullying is present in their life”

-> Perhaps the Author(s) could reformulate this sentence to indicate more clearly the relationship between gifted students and bullying. Are they bullied more?

“However, this model was different for gifted and unidentified students.”

->Instead of suggesting a difference between these groups, the Author(s) should describe the difference here. Especially as the the comparison between these groups is in the main focus of this ms.

Lines 30-31: “Subjective well-being (SWB) is a basic component of health [1] that predicts emotional, cognitive and behavioral engagement, as well as academic performance [2].“

-> I would add here that SWB predicts students’ engagement.

37-38: “In Spain, gifted students scored higher than the general population on levels of bullying and cyberbullying.“

-> The Author(s) should state hare more clearly the source. Scored higher in what?

39: “Researchers have found severe psychological involvement in cases of victimization“

-> Severe psychological consequences, or impact?

110: “Hypothesis 1 (H1): Gifted children will have lower SWB than unidentified children.“

-> The Author(s) could offer a bit more justification for this hypothesis. How did the Author(s) become to hypothesise this difference? Based on the reviewed literature, other kind of hypothesis could have been stated as well, like in the case of Hypothesis 2.

 154: “Hypothesis 3 (H3): EI predicts SWB.“ 

-> The Author(s) should reformulate this hypothesis. It should state explicitly how EI is expected to correlate with SWB (positively or negatively).

1.4. Emotional well-being, emotional intelligence and moods in gifted children and adolescents. 

-> There is some repetition in this section. Some material here (lines 210-215) would be more suitable in the context of hypothesis 1. The most logical place for the hypothesis 5 and the corresponding text, in turn, could be immediately after the Hypothesis 4. Perhaps the Author(s) should consider whether there is a need for this separate section or should it be integrated in those sections.

210-212: “Compared to their peers, they had higher average NA and lower average PA, obtaining a worse result; only the sadness variable was significant.“

-> The Author(s) might want to reformulate this sentence to enhance the articulation. Was it that no differences was found between the gifted students and their peers except in the case of sadness?

212-215: “The fact that many gifted children do not receive an educational response tailored to their needs, coupled with their involvement in harassment or stigmatization situations and social isolation, leads us to believe that, in our country, this population may suffer negative emotions more often than 214 unidentified students.“

-> This sentence should provide references to support the argumentation. Here and elsewhere the Author(s) should use the term Spain instead of “our country”.

Hypothesis 5: Here again, the hypothesis should be stated in more explicit way. What kind of association is expected exactly? The positive indirect effect to be greater in the gifted sample? The justification for this Hypothesis 5 should be articulated more clearly as well. 

2. Materials and Methods (234-245)

-> An error of some kind happened in the submission process, I assume. This text appears to be from a guideline to authors. Please revise.

2.1. Sample and data collection 

-> The Author(s) should provide more information here. How was the gifted students sample determined? What was the response rate for the surveys? Were there any missing observations? Both gifted and unidentified samples, for example, seem to consist mainly of male students. Perhaps males were more willing to participate?

Data analysis

287: “First, descriptive statistics (the mean and standard deviation) were developed“

-> Calculated

No cut-off value for the Root Mean Square Error of Approximation is determined. 

Perhaps the Author(s) could justify their decision not to use any control variables in their analyses.

Results (305-308): 

-> Here again, an error of some kind happened in the submission process. Please revise.

3.2. Test of Hypothesized Model 

->According to the Table 3, the hypothesised model does not seem to fit the data very well. The range of RMSEA values is .0.19-0.31. Hu & Bentler (1999), for example, suggest a cut-off criteria of 0.06 for RMSEA.

-> The indirect effect between EI and SWB (H4 and H5) is not tested and no coefficient is reported. 

-> The results section should be revised to communicate the results more clearly. What paths where significant and what was the effect size. 

357-358: “Moreover, there is a direct effect between EI and SWB in unidentified students but not in gifted students. Thus, H5 is confirmed.“

-> I don’t think this finding confirms the H5. Please find my comment above concerning the formulation of H5. 

Author Response

Reviewer 1

We want to thank the reviewer for his/her constructive feedback and valuable comments. In response to these comments we have made the following revisions.

Reviewer 1 comment: Moderate English changes required

Authors’ response: Thank you very much for your comment. The manuscript has been sent to a translation company specialized on scientific articles, so an expert on the study field has revised the text according to a higher quality English (AJE-American Journal Experts).

Reviewer 1 comment: First, the formulations of the stated hypotheses are not explicit enough (please more detailed comments below).

Authors’ response: Thank you for your suggestion. We will answer now each of the mentioned specific comments. Changes can be seen on the following pages. We have changed the paper, following your advices.

Reviewer 1 comment: Second, a more serious, concern is the conducted analysis and the results section. According to the fit statistics the fit between the theoretical model and the data seems poor (here again, please see my more detailed comments on the RMSEA below).

Authors’ response: Thank you for your comments. We will improve the method and results sections following your comments. We described on the following pages these changes.

Reviewer 1 comment: The results section should also review the findings in more structured and explicit way.

Authors’ response: Following the line of reasoning of the reviewer 1, we have improved result section. We hope this section has a higher quality now.

Reviewer 1 comment: In addition, the Author(s) could reflect a bit more Spain as a national context. Is the some national characteristics that would explain the findings be generalised to other national contexts as well?

Authors’ response: Thank you so much for your suggestions. We have improved the introduction and explained the Spanish context (lines 53-68).

“In Spain, the achievement results in international studies, in comparison with other countries, point to an excessive number of students with low achievement and a small number of students with high achievement [16]. Number of students identified is very low (0.2%) in Spain [17]. The incidence of identified students is very low (0.2%) in Spain. The current Education Act [18] considers that these students may present specific needs for educational support and establishes enrichment and acceleration as measures for their attention. To activate these measures, a favorable report from the authorized School Guidance Service is required. In this sense, in the Valencian Community there are only 94 registered gifted students, and no data for later years [17]. In addition, teachers have little knowledge on the subject, even if they have completed specific training on giftedness [19]. This means that students are not sufficiently stimulated at school and see their underdeveloped potential [16]. As in other countries, gifted Students scored higher than the general population on levels of bullying and cyberbullying. A total of 55.1% of gifted students are victims, but if the victim status is added to that of aggressor, the total is 83.2% of the sample [20]. Specifically, González-Cabrera et al [21] showed that 31.5% were related to the cybervictim profile and 10.6% to the cyberbully profile in a Spanish sample [21]. Almost 25% of respondents believe that, in some way, teachers had fostered them to be victims. Researchers have found psychological impact in cases of victimization (e.g., higher scores of depression, stress and anxiety, as well as lower perceived quality of life) [20].”

“These characteristics can be generalized other countries and contexts (Line 14)”

 “Gifted students are victims of bullying and cyberbullying in a significant percentage [9,10]”

Reviewer 1 comment: There are also some problems with the structure of the ms. and it would benefit from a thorough language check.

Abstract: “Some researchers consider these groups to be vulnerable, and there are studies demonstrating that bullying is present in their life” Perhaps the Author(s) could reformulate this sentence to indicate more clearly the relationship between gifted students and bullying. Are they bullied more?

Authors’ response: We appreciate theses comment so we have modified the abstract. The sentence and there are studies demonstrating that bullying is present in their life”, has been changed for the following one: “more often they participate in situations of harassment as victims and/or stalkers”. Moreover, in the introduction we have included a paragraph showing concrete data (lines 63-67):

“As in other countries, gifted Students scored higher than the general population on levels of bullying and cyberbullying. 55.1% of gifted students are victims, but if the victim status is added to that of aggressor, the total is 83.2% of the sample [20]. Specifically, González-Cabrera et al (2019) showed that 31.5% was related to the cybervictim profile and 10.6% to the cyberbully profile in Spain sample [21].”

As well we have increased the number of references (lines 41-53):

“Gifted children can be seen as different, since they are intellectually more advanced than their classmates and can claim the attention of teachers [10]. In society there is a cultural tradition of anti-intellectualism that often reflects the circumstances of the classrooms [9]. High achievement can be valued by parents and teachers, but not by the general school culture [11]. In addition, these children may experience asynchronous development in social and emotional áreas [12]. Stalkers may include classmates, but also adults (parents and teachers, e.g.). Studies point to gender differences; men are harassed to a greater extent than women [11], but gifted females had more peer victim levels [13]. The strong awareness of justice presented by gifted children can lead them to not understand why something unfair is happening to them if they have done nothing wrong [14] and they may feel confused and distressed [15]. They can live as a failure not being able to solve their problems without the help of their parents or teachers [14] or have anxiety when thinking they are causing them problems and worries [15]. These children are also at risk of becoming bullies in an attempt to regain control [11]. “

Reviewer 1 comment: “However, this model was different for gifted and unidentified students.” Instead of suggesting a difference between these groups, the Author(s) should describe the difference here. Especially as the the comparison between these groups is in the main focus of this ms.

Authors’ response: Thanks again you for your suggestions. Following your proposal, we have modified the sentence (lines 23-24): “The mediating role of positive mood is given in the two groups; however, negative mood only mediates this relationship in gifted students”.

Reviewer 1 comment: Lines 30-31: “Subjective well-being (SWB) is a basic component of health [1] that predicts emotional, cognitive and behavioral engagement, as well as academic performance [2].“ I would add here that SWB predicts students’ engagement.

Authors’ response: Thank you for your comment. Following it, we have included the sentence (lines 34-35): “ SWB predicts student engagement [3,4,5]”

Reviewer 1 comment: 37-38: “In Spain, gifted students scored higher than the general population on levels of bullying and cyberbullying.“ The Author(s) should state hare more clearly the source. Scored higher in what?

Authors’ response: Thank you for your suggestion. Given your comment, we have included more information and have clarified the source (line 64-67): « 55.1% of gifted students are victims, but if the victim status is added to that of aggressor, the total is 83.2% of the sample [20]. Specifically, González-Cabrera et al (2019) showed that 31.5% was related to the cybervictim profile and 10.6% to the cyberbully profile in Spain sample [21]. »

Reviewer 1 comment: 39: “Researchers have found severe psychological involvement in cases of victimization“ Severe psychological consequences, or impact?

Authors’ response: We appreciate this comment, so we have eliminated “Researchers have found severe psychological involvement in cases of victimization”, replacing it by the following one: “Researchers have found psychological impact in cases of victimization”

Reviewer 1 comment: 110: “Hypothesis 1 (H1): Gifted children will have lower SWB than unidentified children.“ The Author(s) could offer a bit more justification for this hypothesis. How did the Author(s) become to hypothesise this difference? Based on the reviewed literature, other kind of hypothesis could have been stated as well, like in the case of Hypothesis 2.

Authors’ response: We highly value your contribution for what we have justified Hypothesis 1, adding the following paragraph (lines 161-166): “There are low identification rates in Spain. These children are identified as different and stigmatized. In addition, there is no adjustment between their characteristics and educational responses, they participate more in situations of bullying and cyberbullying, and most do not feel supported by their teachers. This may be the cause of higher levels of NA and lower levels of AP and, therefore, lower subjective well-being. This allows us to formulate the 1st Hypothesis.”

Reviewer 1 comment: 154: “Hypothesis 3 (H3): EI predicts SWB.“ The Author(s) should reformulate this hypothesis. It should state explicitly how EI is expected to correlate with SWB (positively or  negatively).

Authors’ response: Taking into consideration your suggestion we have changed Hypothesis 3 (line 187) as follow: “Hypothesis 3 (H3): EI positively predicts SWB”.

Reviewer 1 comment: Emotional well-being, emotional intelligence and moods in gifted children and adolescents. There is some repetition in this section. Some material here (lines  210-215) would be more suitable in the context of hypothesis 1. The most logical place for the hypothesis 5 and the corresponding text, in turn, could be immediately after the Hypothesis 4. Perhaps the Author(s) should consider whether there is a need for this separate section or should it be integrated in those sections.

Authors’ response: Thanks for your observation. We have eliminated this paragraph and have improved the context of Hypothesis 1: “In school contexts, many gifted children do not receive an educational response tailored to their needs. The fact, a lack of support and lack of an educational response and it is possible that can lead to the emergence of disruptive behaviors, including challenging teachers’ authority; boredom; and difficulties in relationships with their peers and families [29]”. Moreover, we have added subsection four and five and we have changed its title: “Subjective well-being, emotional intelligence and mood in gifted/unidentified children and adolescents”

Reviewer 1 comment: 210-212: “Compared to their peers, they had higher average NA and lower average PA, obtaining a worse result; only the sadness variable was significant.“ The Author(s) might want to reformulate this sentence to enhance the articulation. Was it that no differences was found between the gifted students and their peers except in the case of sadness?

Authors’ response: Given your suggestion we have reformulated the sentence: “Compared to their peers, they had higher average NA and lower average PA, obtaining a worse result; no significant differences was found between the gifted students and their peers except in the case of sadness” (lines 667-668).

Reviewer 1 comment: 212-215: “The fact that many gifted children do not receive an educational response tailored to their needs, coupled with their involvement in harassment or stigmatization situations and social isolation, leads us to believe that, in our country, this population may suffer negative emotions more often than 214 unidentified students.“ This sentence should provide references to support the argumentation. Here and elsewhere the Author(s) should use the term Spain instead of “our country”.

Authors’ response: We appreciate theses comment so we have completed the sentence with two references and replaced in the whole text “our country” by “Spain” (lines 245-248): The fact that many gifted children do not receive an educational response tailored to their needs, coupled with their involvement in harassment situations, leads us to believe that, in Spain, this population may suffer negative emotions more often than unidentified students.

Reviewer 1 comment: Hypothesis 5: Here again, the hypothesis should be stated in more explicit way. What kind of association is expected exactly? The positive indirect effect to be greater in the gifted sample? The justification for this Hypothesis 5 should be articulated more clearly as well.

Authors’ response: Thank you for your proposal. In the manuscript, the formulation of hypothesis 5 can be a bit ambiguous and lead to different interpretations. Therefore, we have reformulated it following your proposal (lines 263-266): Hypothesis 5 (H5): Moods will differentially mediate the relationship between EI and SWB in gifted and unidentified children and adolescents. In AC students, positive and negative mood mediates the effect of EI on SWB. In NAC students, positive mood mediates the influence of EI on SWB.”

Reviewer 1 comment: Materials and Methods (234-245). An error of some kind happened in the submission process, I assume. This text appears to be from a guideline to authors. Please  revise.

Authors’ response: Thank you for your clarification. We have removed those paragraphs.

Reviewer 1 comment: Sample and data collection. The Author(s) should provide more information here. How was the gifted students sample determined? What was the response rate for the surveys? Were there any missing observations? Both gifted and unidentified samples, for example, seem to consist mainly of male students. Perhaps males were more willing to participate?

Authors’ response: Thank you very much for your comments. We have tried to respond to all of them. Following your suggestions, we have worked to provide a better presentation of the study (lines 269-292): “The study design was cross-sectional and we used the convenience sampling technique [89] based on participants who are easily accessible, but we deliberately choose certain people based on their characteristics. We used this technique because it was necessary to select an equivalent number of gifted and unidentified students with similar characteristics. Moreover, in Spain, the percentage of gifted boys and girls is different; the Ministerio de Educación y Formación Profesional [17] showed that there are more boys identified as gifted (approximately 70%). There are fewer girls because they hide their abilities, and in the classroom, they are called invisible girls in the classroom [90]. The conditions for participating in the study were that the students must be identified or unidentified as gifted and be in compulsory education. Data were collected through self-report questionnaires completed voluntarily by the participants in the presence of the researcher after they had provided their own and their parents’ informed consent. Participants received instructions and information about the procedure for filling out the questionnaire, and the researcher helped to explain that there were no right or wrong answers and was present to resolve any doubts that arose [91]. There were missing data on conflictive items representing approximately 0.2%, and these participants were excluded from the analyses. This study was carried out in accordance with the ethical guidelines of the American Psychological Association and the Declaration of Helsinki, and it received approval from the Ethics Committee of the Catholic University of Valencia (UCV/2015-2016/05).”.

We contacted with the associations, which signed a document giving their acceptance on taking part of the study and guarantying the conditions of suitability of their installations to fill in the questionnaires. We also sent a letter to all its members inviting them to an informative talk. The attendees got answers to all their questions and sign the required authorizations to participate on the study. They were explained the criteria for completing the questionnaires.

Unfortunately, we do not have the required information about the response rate for the surveys. We do not know how many members have each association that meet the inclusion criteria for this study. However, of the participants we contacted individually, the response rate was 100%.

Reviewer 1 comment: 287: “First, descriptive statistics (the mean and standard deviation) were developed“. Calculated

Authors’ response: Thank your appreciation. We have changed the wrong word.

Reviewer 1 comment: No cut-off value for the Root Mean Square Error of Approximation is determined.

Test of Hypothesized Model According to the Table 3, the hypothesised model does not seem to fit the data very well. The range of RMSEA values is .0.19-0.31. Hu & Bentler (1999), for example, suggest a cut-off criteria of 0.06 for RMSEA.

Authors’ response: We appreciate and understand this comment. However, latest researches about fit index show that conventional index and cutoff have limitations. Therefore, following your suggestion, we have improved the data analysis section explaining this (lines 332-344 ): “Since 1990, recommended cutoff values were indices of ≥.90 [100], but it has since been shown that the cutoffs recommended by Hu and Bentler are not universally applicable [101]. Moreover, parameter estimates are influenced by sample size [102]. Additionaly, the root mean square error of approximation (RMSEA) was calculated. The RMSEA is a measure of the average size of the fitted residuals per degree of freedom. Values of approximately .05 or less indicate a close model fit, while values of approximately .08 or less indicate a fair model fit [103]. This fit indicates that the hypothesized relationship in the model are plausible. Despite the popularity of RMSEA, it is not an appropriate fit index in models with small df and/or low Kenny, Kaniskan, and McCoach [104]  showed that in these situations RMSEA have large values and,often, falsely indicated a poor model fit. In fact, Taasoobshirazi and Wang [105] affirm that when sample sizes are smaller than 200, particularly when combined with small degrees of freedom, researchers should avoid reporting RMSEA. Thus, in this research we did not consider RMSEA. If the initial model offers a poor fit to the data, the second step is to modify the model.”

Reviewer 1 comment: Perhaps the Author(s) could justify their decision not to use any control variables in their analyses.

Authors’ response: Thank you very much for your proposal about control variables. The use of the control variables is common in mediation or moderation analysis. In this sense, we previously performed T-test and ANOVA to test if it was necessary to use age, sex and educational level as control variables. As there were no significant differences, we have not shown these results in the first version of the paper. However, after the reviewer’s comments, we have included this information in the paper (lines 317-318; 369-371): “Moreover, two control variables were measured: sex and educational level (primary, compulsory secondary education, vocational training, and high education).“ and “Moreover, in the two samples in this research, there were no statistically significant differences in the study variables based on sex and educational level. Therefore, these control variables were not included in the model.”

Reviewer 1 comment: Results (305-308). Here again, an error of some kind happened in the submission process. Please revise

Authors’ response: Thank you for your comment. We have eliminated the paragraphs.

Reviewer 1 comment: The indirect effect between EI and SWB (H4 and H5) is not tested and no coefficient is reported.

The results section should be revised to communicate the results more clearly. What paths where significant and what was the effect size.

Authors’ response: Thank you for your suggestion. You are right, we did not comment the indirect effect, so we have improved data analysis and results sections (lines 345-354; 388-398; 421-437):

Moreover, we used analytical techniques to analyze direct and indirect effects between variables. The indirect effects involved in the model were tested using method proposed by MacKinnon et al. [106]. Moreover, we analyzed effect size for mediation through complete or partial mediation [107,108].

Combinations of descriptive and inferential statistics were calculated with the Statistical Package for Social the Sciences (SPSS) version 24, SEM was performed using Analysis of Moment Structures (AMOS) software version 24, and PROCESS by Hayes [109]. Bootstrap resampling was performed with confidence intervals set at 95%. The sample size was adequate, the correlations between variables were not high (i.e., 0.85) [110], and the sample data did not follow a standard normal distribution [111].”

To further verify mood’s mediation effect, PROCESS was used and we have computed the mediator’s BC confidence interval. EI is significantly related to positive (Coeff=0.2, SE=0.03, р=0.001) and negative mood (Coeff=-0.1, SE=0.03, р=0.001). Positive (Coeff=1.65, SE=0.17, р=0.001) and negative mood (Coeff=-1.66, SE=0.17, р=0.001) is significantly related to SWB after controlling for EI. The analysis of the indirect effect of EI SWB, through the positive (indirect effect=0.32; 95% BC confidence intervals being [0.19, 0.49]) and negative mood (indirect effect=0.16; 95% BC confidence intervals being [0.05, 0.27]), showed a statistically significant mediation. The direct effect is statistically significant (Coeff=0.26, SE=0.08, р=0.01); thus, we could conclude that positive and negative mood partially mediate the relationship between EI and SWB. These results provide support for H3: EI predicts well-being. The hypothesized mediating role of positive and negative mood in the relationship between EI and SWB was tested, and the results are in support of H4 (Figure 2).”

“The estimators also indicate that all the estimated variables correlate adequately, as well. In the reespecified unidentified student model, EI was significantly related to positive (Coeff=0.18, SE=0.04, р=0.001) but it was not to negative mood (Coeff=-0.04, SE=0.04, р=0.27). Both positive mood (Coeff=1.43, SE=0.24, р=0.001) and negative mood (Coeff=-1.65, SE=0.25, р=0.001) are significantly related to SWB after controlling for EI. The analysis of the indirect effect of EI SWB, through the positive mood (indirect effect=0.26; 95% BC confidence intervals being [0.09, 0.46]) showed a statistically significant mediation. The direct effect is statistically significant (Coeff=0.32, SE=0.12, р=0.006); thus, we could conclude that positive mood partially mediates the relationship between EI and SWB.

In the reespecified gifted students model EI was significantly related to positive (Coeff=0.20, SE=0.05, р=0.001) and negative mood (Coeff=-0.14, SE=0.05, р=0.01). Both positive mood (Coeff=1.84, SE=0.24, р=0.001) and negative mood (Coeff=-1.68, SE=0.23, р=0.001) are significantly related to SWB after controlling for EI. The analysis of the indirect effect of EI on SWB by positive mood (indirect effect=0.37; 95% BC confidence intervals being [0.18, 0.63]) and negative mood (indirect effect=0.24; 95% BC confidence intervals being [0.08, 0.41]) showed a statistically significant mediation. The direct effect was not statistically significant (Coeff=0.18, SE=0.13, р=0.18); thus, we could conclude that positive and negative mood totally mediate the relationship between EI and SWB..”

Reviewer 1 comment:

357-358: “Moreover, there is a direct effect between EI and SWB in unidentified students but not in gifted students. Thus, H5 is  confirmed.“ I don’t think this finding confirms the H5. Please find my comment above concerning the formulation of H5.

Authors’ response: Thank you for your comment. We have revised and modified the formulation of H5 and we have improved result and discussion sections to show how we analyzed H5.

We hope these revisions improve the manuscript and that you and the reviewers will now deem it worthy of publication in the International Journal of Environmental Research and Public Health.

Reviewer 2 Report

Dear Authors,

I think that your paper present an original research which offer new knowledge to the field of subjective well-being. The manuscript subsequently contains new information and I really appreciated to read it. 

few remarks: 

- Regarding methods: I don't undestand the first part of the Material and Method. Is it necessary?

- Since this is a convenince sample  I would prefer to talk about participants; and I think that could be useful a reflection on gender difference (there are no references to this issue  in literature either). 

- pg. 7 line 319 there is a misprint

Author Response

REVIEWER 2

We appreciate the comments received to improve our work, and we have made changes in the manuscript, trying to respond to all these aspects.

Reviewer 2 comment: Regarding methods: I don't understand the first part of the Material and Method. Is it necessary?

Authors’ response: Thank you very much for these comments. You are right, so we have eliminated the sentence.

Reviewer 2 comment: Since this is a convenience sample I would prefer to talk about participants; and I think that could be useful a reflection on gender difference (there are no references to this issue  in literature either).

Authors’ response: Thank you very much for your suggestion. We have changed the title: “Participants and Procedure”.

Moreover, we have improved the material and methods section and we have included gender differences (lines 272-275): “Moreover, in Spain, the percentage of gifted boys and girls is different; the Ministerio de Educación y Formación Profesional [17] showed that there are more boys identified as gifted (approximately 70%). There are fewer girls because they hide their abilities, and in the classroom, they are called invisible girls in the classroom [90].”

Too, we have analyzed gender as control variable (lines: 317-318; 369-371): “Moreover, two control variables were measured: sex and educational level (primary, compulsory secondary education, vocational training, and higher education)” and “Moreover, in the two samples in this research, there were no statistically significant differences in the study variables based on sex and educational level. Therefore, these control variables were not included in the model.

We have also considered the gender differences on the introduction when talking about scholar harassment (lines 46-47):

« Studies point to gender differences; men are harassed to a greater extent than women [11], but gifted females have more peer victim levels [13]. « 

Reviewer 2 comment: pg. 7 line 319 there is a misprint

Authors’ response: Thank you very much for your clarification. You are right, so we have eliminated the misprint.

We hope these revisions improve the manuscript and that you and the reviewers will now deem it worthy of publication in the International Journal of Environmental Research and Public Health.

Round 2

Reviewer 1 Report

Review comments

I appreciate the effort the authors have made in replying to my concerns and comments. I think that the paper is more solid after the revisions and it reads better now. However, there are still some minor concerns I would like to raise.

Abstract:

“These characteristics can be generalized to other countries and contexts.”

No countries or contexts are specified yet in the abstract. Do the Authors mean that these findings have been reported cross-nationally?  Or that the findings of this study can be generalized? In that case, the finding should be reported in the abstract before reflecting on their generalizability. This claimed generalizability should also be arguments more explicitly in the paper.

Introduction:

42-43: “In society, there is a cultural tradition of anti-intellectualism that often reflects the circumstances of the classrooms [9].”

In Spanish society?

I recognize and appreciate the fact that the Authors have now added information on Spain as a national context for this study. However, I would like to see more discussion on whether the results can be generalized to different national context as well. Or are they characteristic for the Spanish education system. Perhaps the authors could briefly reflect this issue in the limitations section.

Hypotheses:

I appreciate that the Authors have added more justification for their hypotheses. However, I still have a couple of concerns:

First (and more minor one), the Authors should add references when explaining their first hypothesis (“There are low identification rates in Spain. These children are identified as different and stigmatized. In addition, there is no adjustment between their characteristics and educational responses, they participate more in situations of bullying and cyberbullying, and most do not feel supported by their teachers. This may be the cause of higher levels of NAand lower levels of AP and, therefore, lower SWB. This allows us to..“).

Second, I still don’t see quite sufficient argumentation in the literature review to support the H5: “In AC students, positive and negative mood mediates the effect of EI on SWB. In NAC students, positive mood mediates the influence of EI on SWB.“

The hypothesized mechanism should be described more explicitly here. Is it that, theoretically, EI should not be connected to negative mood for unidentified students? Or that negative moods would not be correlated with SWB for unidentified students? Due to the lack of clarity here, this part of the analysis seem a bit exploratory. Perhaps the Authors could consider removing this separate hypothesis (H5) and simply adding to the H4 that the hypothesized mediation will be tested separately for gifted and unidentified students.

In addition, I did not find the exact sentence in the text where the Authors introduce the acronyms of AC and NAC. These acronyms are not used in case of other hypotheses.

Methods:

“The study design was cross-sectional and we used the convenience sampling technique [89] based on participants who are easily accessible, but we deliberately choose certain people based on their characteristics. We used this technique because it was necessary to select an equivalent number of gifted and unidentified students with similar characteristics.“

The authors could add more details on how exactly were the two comparable samples collected. This sounds a bit like matching procedure without using analytical tools. How where the unidentified students chosen to find “unidentified students with similar characteristics [than the gifted students]”? Were they on a same class than the gifted students? Or were they sampled from different schools and / or classes? How many classrooms and schools were included in the final samples?  The authors should also state in the manuscript that the exact response rate could not be determined.

The Authors have now provided justification for their choice of utilized fit indices. Perhaps the Authors could consider removing all the RMSEA estimates from the tables as they have decided not to consider them in the model fit analysis. Or, if they are still presented in the tables, they should be reference in the text as well.

“Bootstrap resampling was performed with confidence intervals set at 95%.“ How many replications were run in the bootstrap analysis?

3.2. Test of the Hypothesized Model

Results section is not the most suitable place for methodological details and references. I would remove this discussion to the methods section and report only empirical findings in the results section.

I am not a native English speaker myself but I still believe that the paper would benefit from a spell check.

For example (line 392): “The analysis of the indirect effect of EI SWB”. Did the Authors mean indirect effect of EI on SWB?

AND

Lines 339-340: “Despite the popularity of RMSEA, it is not an appropriate fit index in models with small df and/or low Kenny, Kaniskan, and McCoach [104]..” This seems to be lacking some information. Low what?

Author Response

REVIEWER 1

Thank you for the opportunity to revise our manuscript “Subjective emotional well-being, emotional intelligence, and mood of gifted vs unidentified students: a relationship model”. We want to thank the reviewer for his/her constructive feedback and valuable comments. In response to these observations we have made the following changes.

Kind regards,

The authors

Reviewer 1 comment: Moderate English changes required

Authors’ response: Thank you very much for your comment. The manuscript has been sent again to a professional native English speaker to review the possible minor spell check required. We attach two certificates.

Reviewer 1 comment: Abstract: “These characteristics can be generalized to other countries and contexts.”

No countries or contexts are specified yet in the abstract. Do the Authors mean that these findings have been reported cross-nationally?  Or that the findings of this study can be generalized? In that case, the finding should be reported in the abstract before reflecting on their generalizability. This claimed generalizability should also be arguments more explicitly in the paper.

Authors’ response: Thank you very much for your contribution. We have deleted this sentence.

Reviewer 1 comment: Introduction: 42-43: “In society, there is a cultural tradition of anti-intellectualism that often reflects the circumstances of the classrooms [9].” In Spanish society?

 I recognize and appreciate the fact that the Authors have now added information on Spain as a national context for this study. However, I would like to see more discussion on whether the results can be generalized to different national context as well. Or are they characteristic for the Spanish education system. Perhaps the authors could briefly reflect this issue in the limitations section.

Authors’ response: We greatly appreciate your suggestion and have included it (line 63): ‘In Spanish Society”. We have included one more limitation (line 1267-1273):  “Fourth, the topics analysed by our study (SWB, EI and mood) are susceptible to cultural differences (e.g. [133-138]). However, there are also cultural variations about the concept of gifted children [139], gifted education [140]...  Literature suggests that culture is one of the most important sources of psychosocial analysis but personality, socioeconomic class, education, sex, age, … are also of particular relevance. Moreover, some authors report cultural mismatch between teachers and students [141]. Following Chowdhury [142], futures studies should analyse the diversity of peoples and their cultures in the era of globalization.”

Bagheri, Z., Kosnin, A.M., Besharat, M.A. (2013). The influence of culture on the functioning of emotional intelligence. 2nd International Seminar on Quality and Affordable Education (ISQAE 2013), 2013. Available from: https://educ.utm.my/lv/wp-content/uploads/2013/11/181.pdf

Nayak, M. (2014). Impact of culture linked gender and age on emotional intelligence of higher secondary school adolescents. International Journal of Advancements in Research & Technology, Volume 3, Issue 9, September -2014.

Carballeira, M., González, J.A.; Marraro, R.J. (2015). Diferencias transculturales en bienestar subjetivo: México y España. Anales de Psicología, 2015, 31(1), 199-206. Doi: 10-6018/analesps.31.1.166931

Tov, W., & Diener, E. (2007). Culture and subjective well-being. In S. Kitayama & D. Cohen (Eds.), Handbook of cultural psychology (pp. 691-713). New York: Guilford

Luomala, Harri & Kumar, Rajesh & Worm, Verner & Singh, JD. (2004). Cross Cultural Differences in Mood Regulation: An Empirical Comparison of Individualistic and Collectivistic Cultures. Journal of International Consumer Marketing. 16. 39 -62.

Watson, D., Clark, L. A., & Tellegen, A. (1984). Cross-cultural convergence in the structure of mood: A Japanese replication and a comparison with U.S. findings.

Freeman, J. (2015), ‘Cultural Variations in Ideas of Gifts and Talents With Special Regard to the Eastern and Western Worlds’, in Dai, D.Y. & Ching, C.K. (Eds.) pp. 231-244. Gifted Education in Asia. Charlotte, NC: Information Age Publishing

Ford, D.Y., Moore, J.L., Milner, H.R. (2015). Beyond cultureblindness: A model of culture with implications for gifted education. Roeper Review 27(2):97-103

Chowdhury, M.A. (2017). Towards the achievement of a unified, uniform and socially-just ‘gifted education’ policy acceptable on a global scale. Journal for the Education of Gifted Young Scientists, 5(1), 1-22. http://dx.doi.org/10.17478/JEGYS.2017.51

Ford, D.Y. (2010). Culturally Responsive Classrooms: Affirming Culturally Different Gifted Students. Multiculutral Issues, 33(1), 50-53. Available from: https://files.eric.ed.gov/fulltext/EJ874024.pdf

Reviewer 1 comment: Hypotheses: First (and more minor one), the Authors should add references when explaining their first hypothesis (“There are low identification rates in Spain. These children are identified as different and stigmatized. In addition, there is no adjustment between their characteristics and educational responses, they participate more in situations of bullying and cyberbullying, and most do not feel supported by their teachers. This may be the cause of higher levels of NAand lower levels of AP and, therefore, lower SWB. This allows us to..“).

Authors’ response: Thank you for your comment. Following your proposal, we have modified the text (line 301-304): “There are low identification rates in Spain [17]. These children are regarded as different and stigmatized [10]. In addition, there is no adjustment between their characteristics and the educational responses they receive [16], they are more involved in situations of bullying and cyberbullying [21], and most do not feel supported by their teachers [20]. This may be the cause of higher levels of NA and lower levels of AP and, therefore, lower SWB [25]. This allows us to..“).”

Reviewer 1 comment: Hypotheses: Second, I still don’t see quite sufficient argumentation in the literature review to support the H5: “In AC students, positive and negative mood mediates the effect of EI on SWB. In NAC students, positive mood mediates the influence of EI on SWB.“

The hypothesized mechanism should be described more explicitly here. Is it that, theoretically, EI should not be connected to negative mood for unidentified students? Or that negative moods would not be correlated with SWB for unidentified students? Due to the lack of clarity here, this part of the analysis seem a bit exploratory. Perhaps the Authors could consider removing this separate hypothesis (H5) and simply adding to the H4 that the hypothesized mediation will be tested separately for gifted and unidentified students.

Authors’ response: Taking into account your valuable contribution, we have removed hypothesis 5 and included the following sentence (line …): “Based on this information,  hypothetical mediation (H4) will be tested separately for gifted and unidentified students. Then, in AC students, positive and negative mood will mediate the effect of EI on SWB and in NAC students, positive mood will mediate the influence of EI on SWB.”

Reviewer 1 comment: In addition, I did not find the exact sentence in the text where the Authors introduce the acronyms of AC and NAC. These acronyms are not used in case of other hypotheses.

Authors’ response: Thank you very much for your comment. We made a mistake, so we have included the acronym in the abstract (line 12,19).

Reviewer 1 comment: “The study design was cross-sectional and we used the convenience sampling technique [89] based on participants who are easily accessible, but we deliberately choose certain people based on their characteristics. We used this technique because it was necessary to select an equivalent number of gifted and unidentified students with similar characteristics.“

The authors could add more details on how exactly were the two comparable samples collected. This sounds a bit like matching procedure without using analytical tools. How where the unidentified students chosen to find “unidentified students with similar characteristics [than the gifted students]”? Were they on a same class than the gifted students? Or were they sampled from different schools and / or classes? How many classrooms and schools were included in the final samples?  The authors should also state in the manuscript that the exact response rate could not be determined.

Authors’ response: We highly appreciate your suggestions. We used the convenience criteria of sampling described by Etikan, Musa, & Alkassim (2016). Taking into consideration the reviewer instructions, we have included the following sentence (lines 665-667) to explain how the samples were collected (line 715-717): “The criteria used to collect both samples were: easy accessibility, geographical proximity, availability at a given time and willingness to participate, that are included for the purpose of the study.” In this sense, we did not take into account the school where the students were studying. Their demographic characteristics were analysed (e.g. sex, age, course studied). Moreover, following the suggestion of the reviewer, we have included in this paper the sentence (line 727) “exact response rate could not be determined”.

Dörnyei, Z. (2007). Research methods in applied linguistics. New York: Oxford University Press

Ilker Etikan, Sulaiman Abubakar Musa, Rukayya Sunusi Alkassim (2016). Comparison of Convenience Sampling and Purposive Sampling. American Journal of Theoretical and Applied Statistics 2016; 5(1): 1-4

Reviewer 1 comment: The Authors have now provided justification for their choice of utilized fit indices. Perhaps the Authors could consider removing all the RMSEA estimates from the tables as they have decided not to consider them in the model fit analysis. Or, if they are still presented in the tables, they should be reference in the text as well.

Authors’ response: Thank you for your comment. Following the reviewer suggestion, we have removed this information.

Reviewer 1 comment: 39: “Bootstrap resampling was performed with confidence intervals set at 95%.“ How many replications were run in the bootstrap analysis?

Authors’ response: Thank you for your contribution. Streukens & Leroi-Werelds (2016) showed that the number of bootstrap replications varies between authors (500-5000).  Following Hesterberg (2015) in the present study the 95% confidence interval of the indirect effects was obtained with 2000 bootstrap resamples. We have introduced this information in the paper (line 857) “ 2000 bootstrap resampling were performed with confidence intervals set at 95%.”

Hesterberg, T.C. (2015). What Teachers Should Know About the Bootstrap: Resampling in the Undergraduate Statistics Curriculum. The American Statistical, 69(4), 371-386. DOI: 10.1080/00031305.2015.1089789

Streukens, S., & Leroi-Werelds, S., Bootstrapping and PLS-SEM: A step-by-step guide to get more out of your bootstrap results, European Management Journal (2016), http://dx.doi.org/10.1016/j.emj.2016.06.003

Reviewer 1 comment: Results section is not the most suitable place for methodological details and references. I would remove this discussion to the methods section and report only empirical findings in the results section.

Authors’ response: Thank you for your comment. Following the reviewer suggestion, we have removed this information: “To test the hypotheses, path analysis was then conducted using the ML estimation method. Given the sample size and the number of parameters to be estimated, we modelled relationships among observed (not latent) variables. We tested a moderated model that included all the study hypotheses. This model proposed the mediating effect of mood in the relationship between EI and SWB. The path analysis approach allows the hypothesized relationships to be examined. This analytical technique facilitates the investigation of direct and indirect effects between variables [77]. The indirect effects involved in the model were tested using the bias-corrected (BC) bootstrap confidence interval method [78].” (line 838-846)

Reviewer 1 comment: I am not a native English speaker myself but I still believe that the paper would benefit from a spell check. For example (line 392): “The analysis of the indirect effect of EI SWB”. Did the Authors mean indirect effect of EI on SWB? AND Lines 339-340: “Despite the popularity of RMSEA, it is not an appropriate fit index in models with small df and/or low Kenny, Kaniskan, and McCoach [104]..” This seems to be lacking some information. Low what?

Authors’ response: The manuscript has been sent again to a professional native English speaker to be reviewed. As this is third time, we attached the three certificates correspondig to the different reviews. Moreover, we have changed the sentence of line 392 and removed the second sentence.

Once again, we appreciate the opportunity to review our work for consideration for publication in International Journal of Environmental Research and Public Health. Thanks to the suggestions of the reviewer. We have worked hard to answer them. We really appreciate these comments and we hope our paper to be now of sufficient quality to be accepted for publication in International Journal of Environmental Research and Public Health.

Kind regards,

The authors
